# Understanding Rumen Microbiology: An Overview

Hunter G. Perez [ID], Claire K. Stevenson, Jeferson M. Lourenco [ID] and Todd R. Callaway *[ID]

Department of Animal and Dairy Science, University of Georgia, 425 River Road, Athens, GA 30602, USA; hunter.perez@uga.edu (H.G.P.); claire.stevenson1@uga.edu (C.K.S.); jefao@uga.edu (J.M.L.)
* Correspondence: todd.callaway@uga.edu

**Definition:** The rumen is the largest of the four chambers of the "stomach" in ruminant animals, which harbors an incredibly dense, diverse, and dynamic microbial community crucial for feedstuff degradation, animal health, and production. The primary objective of this article is to enhance knowledge and comprehension of rumen microbiology by providing an introductory-level overview of the field of rumen microbiology. Ruminants possess a distinctive digestive system optimized for the microbial breakdown of complex plant materials. The ruminant "stomach" consists of four chambers (e.g., reticulum, rumen, omasum, and abomasum), which is home to a microbial population that degrades feedstuffs consumed by ruminant animals. Dr. Robert Hungate and Dr. Marvin Bryant's groundbreaking research in the 1960s laid the foundation for understanding the function of the ruminal microbial ecosystem. Recent advancements (e.g., next-generation sequencing) have provided the field with deeper insight into populations, boosting our understanding of how the microbial population of the rumen functions in a variety of conditions. The ruminal microbial ecosystem is comprised of bacteria, along with archaea, protozoa, bacteriophage, and fungi, each contributing to the symbiotic relationship between the microbial ecosystem and the host animal that is essential for optimal animal health and efficient animal production. Traditional anaerobic growth techniques have facilitated the study of individual anaerobic bacteria but have been limited by dependence on growth in laboratory conditions. The development of 16S rRNA sequencing allows the identification of microbial populations that cannot be grown and allows an unbiased view of microbial diversity. Diet shapes the rumen microbial population composition, influencing animal production metrics such as feed efficiency, methane emissions, and immunological functions. Feed additives (e.g., essential oils, eubiotics) hold promise by manipulating and unraveling the microbial biochemical potential for improving animal health, feed efficiency, environmental impacts, and overall production sustainability. Future research impacts include the development of probiotics, prebiotics, and genetic strategies for optimizing the rumen microbiome's multifaceted impacts.

**Keywords:** rumen microbiology; rumen; microbiology; microbiome

## 1. Introduction

Ruminant animals have a unique digestive system that enables the degradation of plant materials such as cellulose, hemicellulose, and lignin that other animals cannot readily utilize [1,2]. A ruminant's "stomach" is comprised of four chambers: the reticulum, rumen, omasum, and abomasum. The rumen is the largest chamber (50–70 L in cattle) and is the location where ingested plant material is fermented by a resident microbial population to produce volatile fatty acids (VFAs), which provide energy to the host ruminant animal [3]. The rumen is home to a dense and diverse population of microorganisms, including bacteria, archaea, protozoa, fungi, and bacteriophage. The ruminal microbial ecosystem plays an essential role in host animals' health and nutrition [2] but also drives sustainability and environmental impacts of ruminant animal production.

Investigation into rumen function dates back to studies by Aristotle, but controlled scientific research into the rumen microbial activity truly began in the late 19th century [4].

In the 20th century, Robert Hungate and Marvin Bryant were pioneers in understanding the rumen microbial ecology in cattle [2,5]. The rumen microbial population is a sustainable ecosystem due to its constant and dynamic nature [2,6]. Cellulolytic bacteria (those that can break down cellulose) are found within the rumen, and their populations can be counted and activities examined using an anaerobic and reduced roll tube technique used to culture strictly anaerobic bacteria (bacteria that do not utilize oxygen and may even be killed by oxygen), which is today known as the "Hungate method" [7]. Microbial growth media have been developed that simulate the environmental conditions (e.g., neutral pH, low redox potential, anaerobic) of the rumen, allowing the cultivation of a wide variety of anaerobic microorganisms, and these have contributed greatly to increasing our understanding of ecological theory.

While research has focused on the function of the microbial ecosystem, progress was limited by the need to grow microbes in liquid broth media or on solid agar media. In fact, it was generally accepted that "to know it, you must grow it", yet many microbes still cannot be cultured and so must be identified at a genetic level (e.g., microbiome). In 2010, the Journal of Animal Science featured a mere four research articles utilizing the term "microbiome", but in 2020, there were more than 184 publications that mentioned "microbiome" [8]. Next-generation sequencing utilizing 16S rRNA has become tremendously less expensive, enabling more researchers to explore the rumen microbial population at a genetic level (microbiome) and allowing a deeper exploration of interactions and dietary effects [9].

## 2. A Glimpse into the Rumen Microbial Ecosystem

Ruminal microbes and the host animal have a mutualistic symbiotic relationship that is critical for both parties [3]. The rumen microbial population is fueled (e.g., energy, nitrogen, and minerals) by ingested feedstuffs that provide nutrients to allow microbial fermentation and growth [2,6]. The rumen is a microbially-friendly environment that serves as a continuous-flow fermentation vat [2]. It produces end-products (e.g., VFAs) absorbed across the ruminal epithelium and utilized for energy by the host animal, and microbial crude protein, which is absorbed across the small intestine as an amino acid source for the animal [3]. The rumen population is very dense (in excess of $10^{10}$ cells/mL of ruminal fluid) and diverse (more than 2500 identified species), but it is important to point out that a substantial proportion (>80%) of the ruminal microbial population remains unknown [9–11]. The rumen microbial population is comprised of bacteria, archaea, protozoa, fungi, and bacteriophage.

Bacteria are the most abundant microbial kingdom within the rumen [1], with solid-associated (adherent) bacteria making 70–80% of the rumen microbial population [12]. The rumen bacterial population is a complex and diverse set of anaerobic bacteria [3], each thriving and competing by utilizing different fermentation substrates through a variety of biochemical/ecological strategies to maximize their competitive success. More than 2500 bacterial species have been isolated from the rumen, and the bacterial population is often in excess of $10^{10}$ cells/mL [3,10]. Because ruminant diets are variable and contain forage and cereal grains, ruminal bacteria must be able to degrade a wide variety of substrates and pivot their metabolic processes rapidly in changing conditions. Some of the most prevalent bacterial species within the rumen are *Ruminococcus*, *Fibrobacter*, *Prevotella*, *Selenomonas*, *Streptococcus*, and *Ruminobacter* [3].

Archaea are more primitive single-celled microbes that exhibit diverse metabolic pathways that can be found in various extreme environments and are often called extremophiles. However, ruminal archaea are exclusively anaerobic methanogens [13], meaning that they utilize hydrogen and carbon dioxide which are produced by other microbial populations to produce methane. In fact, because ruminal archaea have such a high affinity for hydrogen gas, they maintain an extremely low partial pressure of $H_2$ gas, ensuring that the degradation of NADH to NAD in ruminal bacteria can occur spontaneously; thus, methane production acts as a reducing equivalent sink in the rumen. Without ruminal

methane production, the catabolism of feedstuffs would be slowed or stopped due to accumulation of NADH; thus, ruminal methanogens are a critical component of proper rumen function [3,14]. Methane is an important player in the carbon cycle and is a greenhouse gas, and ruminant nutritionists have focused on methane reduction strategies because it also represents a loss of up to 12% of the dietary carbon and energy that the animal consumes. Methanogenic archaea have been observed living on and within ruminal protozoa and on fungal sporangia, also acting as a reducing equivalent sink for these important microbes [15,16].

Protozoa account for 50% of the rumen biomass [17,18], and these motile organisms were recognized early in the history of ruminal microbiology [19]. Protozoa are eukaryotes that can engulf and ferment feedstuffs and also play a crucial role in engulfing bacteria [20]. Rumen protozoa play a pivotal role in rumen metabolism and nutrition, significantly contributing to total volatile fatty acid (VFA) production [20]. Defaunation, or removal of protozoa, can reduce animal performance by approximately 10% [20,21]. Protozoa are difficult to grow in the laboratory, so less is known about their physiology and ecology. Although difficult to grow, protozoa can be studied utilizing optical microscopy analysis and identified using 18S rRNA sequencing [21].

Fungi make up approximately 10% of the total microbial mass of the rumen microbiome. Although, their presence remained largely overlooked until about 35 years ago [22–26]. Ruminant fungi have been known to attach to forage in the rumen, and they act as primary colonizers of forage, initiating the breakdown of fiber [27]. Fungi have a complex life cycle in the competitive ruminal ecosystem, which plays a vital role in fiber degradation, where zoospores initially colonize the plant material and penetrate the surface of the material, secreting degradative enzymes from their fungal rhizobia, breaking open fiber for further microbial attack in a fashion similar to tree roots degrading a concrete sidewalk [28]. As the fungal colony on forage matures, it grows into a large sac-like structure termed a sporangium, which contains daughter zoospores to continue the colonization process. Like protozoa, rumen fungi are closely associated with methanogenic archaea (especially the surface of the sporangium) to aid in $H_2$ utilization and provide support for catabolic biochemical pathways like glycolysis [29]. However, much of the role of fungi in ruminal degradation remains unknown [5].

Bacteriophage are viruses that prey upon bacteria exclusively [30]. These viruses infect bacteria, hijack their replicative machinery, and lead to bacterial cell lysis (resulting in the release of cell contents into the environment). While their exact ecological role in the rumen remains unknown, it is clear that they are involved in diurnal variation in the ruminal bacterial populations, as well as in nutrient cycling (e.g., N recycling).

*Factors Influencing Rumen Microbiota*

Ruminal pH is a critical factor that impacts the digestive processes in ruminant animals. Maintaining a near-neutral rumen pH is crucial for the degradative activity of microbes and nutrient absorption [31]. Rumen pH is typically in the 6–7 range on forage diets but can fall lower when readily fermentable starch is available [32,33]. Changes in ruminal pH can significantly impact microbial function and feedstuff degradation [34]. When pH falls below 6.0, fiber-degrading bacterial populations are inhibited, and forage degradation is reduced. Acute or chronic low rumen pH, referred to as acidosis, can dramatically alter the ruminal microbial ecosystem, leading to a decrease in feed efficiency and reduced growth or milk production [35,36]. Although pH is an important factor that influences the rumen microbiota, there are many other factors that also influence this environment. Host genetics, age, diet, and geographic location can all play a role in influencing the development and establishment of the rumen microbiome [9,37]. Thus, understanding and managing these influential factors is vital to ensuring the efficiency of nutrient utilization in ruminant animals by impacting the microbial population of the rumen.

Overall, the rumen microbial population is a dense, diverse, dynamic, and interlinked ecosystem fueled by the nutrients of the host animal, which creates an environment perfect

for microbial fermentation. The symbiotic relationship between the host animal and the microorganisms within the rumen underscores the importance this environment plays in ruminal function and other biological/ biochemical pathways. Research into this complex environment is rapidly evolving and is critical to expanding our understanding of how the rumen microbial ecosystem impacts nutrition, animal health, and the environment.

## 3. Applied Practices in Rumen Microbiology

### 3.1. Rumen Fluid Collection

Various methods have been developed to collect ruminal fluid, which contains the living anaerobic microorganisms and their metabolic end products. Ruminal bacteria can be categorized as solid-associated (attaches to forage), liquid-associated (thrive in a liquid milieu of rumen), and rumen epithelium-associated (attached to rumen epithelium) [9]. Most commonly, ruminal fluid is collected from cannulated animals, through the use of esophageal tubing or by rumenocentisis [38–40]. Rumen cannulation involves surgical implantation of a cannula (e.g., rubber plug) that provides ready access directly to the rumen [41,42]. The use of ruminal cannulas is somewhat limited based on the cost and invasive nature of cannulation, which tends to limit the number of animals used in studies [39]. Additionally, cannulation allows oxygen flow into the rumen; any oxygen that enters the rumen through the cannula or with feed or water is utilized by facultative anaerobes, which can utilize oxygen but do not require it, but this $O_2$ flow can impact the rumen function [3].

Esophageal tubing is a less invasive alternative to rumen cannulation and is a more practical method when large numbers of animals must be sampled [43,44]. This method involves inserting lubricated vinyl tubing into the mouth of the animal down the esophagus and entering the rumen, and then ruminal fluid is pumped out into a temperature-controlled, gas-impermeable container. However, stomach/esophageal tubing has its limitations as this method does not allow a determination of the exact ruminal location where the ruminal fluid was collected, and it is also more prone to oxygen contamination during collection [45]. Rumenocentesis is a relatively new technique that is used under veterinary supervision where a large gauge needle is inserted into the ventral sac of the rumen through the body wall, and ruminal fluid is withdrawn via syringe [39]. Use of rumenocentesis is somewhat limited due to the difficulty of the process and the risk of producing an abscess or peritonitis, and is also more expensive and difficult to collect from large numbers of cattle. Numerous studies have been conducted comparing these rumen fluid collection methods, and all are useful in specific situations, but each method has specific benefits and limitations [43,46,47].

Once collected, rumen contents are typically filtered through eight layers of cheesecloth or through commercial paint strainers to remove large particulate matter. However, the removal of particulate matter does alter (bias) the microbial population in the remaining ruminal fluid. Further, it is important to utilize glass, metal, or butyl rubber tubing to prevent $O_2$ contamination. Media used to grow ruminal bacteria must be made anoxic (typically via autoclaving followed by flushing with $O_2$-free $CO_2$) and must have a low oxidation–reduction potential (a very reduced redox potential of $-0.4$ V), which can be attained by the use of reducing agents, most commonly used is cysteine [3].

Collection of rumen fluid also allows for its use in in vitro (test tube) studies, which allow researchers to conduct experiments in isolation from the animal-level effects. In vitro studies can be used to determine the rate and amount of $H_2$ and $CH_4$ production, pH changes, $NH_3$ production from protein, and VFA production. In vitro studies can also shed light on the maximal rate of degradation of feedstuffs, including forage and grain.

### 3.2. Counting Microbes

"If you can't grow it, you can't know it" has been the mantra for much of the history of microbiology. The recent advent of molecular biological and next-generation sequencing techniques has reduced the veracity of this statement, but one of the greatest limitations to the use of 16S rRNA microbial analysis in place of growth-based studies is that it cannot

distinguish if the microorganism within the sample are dead or alive [8]. As we navigate the evolving microbiological methodologies, it becomes evident that while molecular techniques offer unprecedented insights, the ability to visualize and grow microbial populations remains a crucial aspect. Thus, it is still important to understand methods to identify live bacterial populations, ensuring a comprehensive and accurate view of the dynamic microbial communities in the rumen microbial ecosystem.

### 3.3. Liquid and Solid Growth

Most rumen bacteria can live in the fluid phase of the rumen, but many can grow on solid material. However, it is relatively difficult to make agar plates that are consistently anaerobic and reduced without the use of anaerobic chambers and including reducing agents in the media [48,49]. The Hungate method for growing anaerobes on solid media involved horizontally rolling the anaerobic (flushed with $O_2$-free $CO_2$) tube in ice to allow for bacterial growth in solid media [7]. The use of isolation/growth techniques like streaking, dilution series, and spread plating [50] allows researchers to obtain a pure bacterial culture from the diverse rumen environment. Other techniques used to grow ruminal bacteria involve the use of highly reduced, anoxic broth (liquid) media. Surprisingly, liquid media are relatively easy to make, sufficiently anoxic, and reduced to support the growth of many ruminal bacterial and archaeal species in pure and mixed cultures in broth cultures. The use of traditional microbial growth-based techniques is limited as we can only grow 10% to 15% of rumen microbes [9,51].

### 3.4. 16S Rrna Sequencing

Ribosomal RNA was first suggested as a tool to gain insights into microorganisms by Olsen and Woese in 1993 [52]. This method of sequencing allows a microscopic perspective on small fragments of DNA, which can provide researchers with a way to understand the diversity and identification of the microorganisms present within the rumen [8]. Next, generation sequencing techniques amplify hypervariable regions of the 16S Rrna gene to allow the identification of the bacterial species in an environment [53–55].

## 4. Contributions to the Scientific Community

Much of our knowledge about microbial ecology and interactions was pioneered in the rumen [3]. As we discover more about the rumen microbial ecosystem, we increase our understanding of the systemic impact that the microbiota has on the host animal. As our understanding of the microbial population has increased, we have developed a new term, "microbiome". The microbiome includes the DNA of all the microorganisms in a sample, living or dead. Through microbiome analysis, we are beginning to understand how the microorganisms that colonize the gastrointestinal tract influence feed efficiency (feed to gain), carcass quality, greenhouse gas emissions, animal health, immune status, and food-borne pathogenic bacterial populations. Understanding these metrics can guide producers in decision-making that impacts their cattle. While we recognize that the microbiome of the rumen can have broad impacts on animal health, it is not clear what constitutes a "good" or a "bad" microbiome.

The diet of a ruminant animal greatly Impacts the rumen microbial population, and using the diet, we can alter the rumen environment. Generally, a high-concentrate diet of cereal grains contains readily fermentable starch, which leads to lactic acid production; because lactic acid is a strong acid, it decreases ruminal PH [35,56]. A decrease in PH reduces microbial diversity and richness [57]. *Streptococcus bovis* is a starch-fermenting ruminal bacteria that thrives in a lower PH environment and is selected for in the rumen of grain-fed cattle by the lower PH [58]. An accumulation of ruminal lactate and a low PH can lead to acidosis. Ruminal acidosis inhibits the forage-degrading microorganisms, resulting in a ruminal population increasingly favorable to starch-degrading species, leading to a "typhooning" of acidosis [59]. Acidosis can lead to keratinization and cracking of the

ruminal wall, leading to peritonitis and liver abscesses, but also to a decrease in feed intake, reduced feed efficiency, and potentially even death of cattle [56,60,61].

Many factors contribute to the feed efficiency of cattle, including genetics, feed type, stress, animal health, climate, and the native ruminal microbiome. Research into the interactions between the rumen and hindgut microbial population has begun to suggest a common microbial population among more feed-efficient animals [62–66]. Cattle producers continuously strive to improve feed efficiency while simultaneously reducing our environmental footprint in terms of fewer days on feed and reduced greenhouse gas emissions (e.g., $CH_4$ and $N_2O$). In order to reduce greenhouse gas emissions, feed additives such as nitrates, lipids, and algae have been used [67–70]. Ionophores have also been used to reduce methane emissions by up to 30–50%, but with increasing concern about antimicrobial resistance, limit the use of ionophores [3,71–73]. By improving feed efficiency, producers can shorten the finishing time period, resulting in fewer days on a feedlot, fewer days on feed and water, less time to potentially get sick or develop antibiotic resistance, less input costs, and younger beef in the processing plant. With, of course, the added benefit of reducing greenhouse gas emissions from beef production.

### 4.1. Foodborne Pathogen Carriage

Among the microorganisms that inhabit the rumen, some are public health concerns. Common foodborne pathogens such as verotoxicytogenic *E. coli* (VTEC), *Salmonella*, and *Campylobacter* thrive within the GIT of many livestock species, including cattle. Each year, more than 9 million people contract a bacterial foodborne illness in the United States each year with over 1351 deaths [74]. A great deal of research has been focused on reducing the population of foodborne pathogenic bacteria in the gastrointestinal tract prior to harvest [75,76]. If pathogen populations entering the processing facility are reduced, then the pathogen reduction strategies interventions utilized during carcass processing will be more effective against a reduced pathogen load. Fecal *E. coli* shedding was decreased when switching from a high-concentrate diet to a high-quality forage diet [77,78]. Feed additives such as sodium chlorate have also been used to decrease the populations of *E. coli* and *Salmonella Typhimurium* during shipment to processing [75,79].

### 4.2. Eubiotics

Direct-fed microbials (or eubiotics) encompass a diverse range of products (e.g., probiotics, prebiotics, and postbiotics) fed to animals to enhance their performance, health, and food safety [80–82]. Probiotics are a subset of eubiotics that are live cultures of lactic acid bacteria, yeasts, or by-products that aim to positively impact the microbial community in the digestive system. Probiotics alter the microbial ecology of the gastrointestinal tract and host animal immune system through a variety of mechanisms [83,84]. Postbiotics are products of the fermentation of yeast or fungal fermentations that are not living but contain metabolic intermediates and end-products [85,86]. Historically, eubiotic approaches have had variable efficacy due to quality control issues as well as a lack of understanding of the modes of action of the individual eubiotic. However, as researchers uncover precise modes of action, understanding has improved, leading to better application and more precise feeding strategies. Producers have utilized eubiotics to boost growth rate, milk production, and overall efficiency, with newer insights revealing additional benefits for cattle performance [81,87,88]. Probiotics are continuously evolving, and the use of DFMs continues to play a vital role in improving the well-being and productivity of animals.

### 5. Conclusions

Rumen microbiology has provided significant contributions to our knowledge of microbial ecology and the interactions between the host animal and the microbial population of the gastrointestinal tract. The rumen is an important key to the future of sustainable animal production, environmental responsibility, and human well-being. Optimizing the rumen microbiome not only enhances animal welfare and productivity but also ad-

dresses environmental impacts linked to livestock production. However, it is essential to acknowledge that there is still much research remaining to be performed. Even though we understand host animal genetics as factors that can influence the rumen microbiome, there remains considerable research to be conducted to understand the intricate interplay between host genetics and the rumen and hindgut microbiome and microbial activities. Furthermore, investigating dietary and microbial shifts throughout an animal's lifetime is essential to comprehending its lasting impact on the rumen microbiome. By understanding these interactions and their effects, the manipulation of the rumen microbiome, particularly through dietary interventions, becomes crucial in addressing concerns related to foodborne pathogens, contributing to improved food safety and overall agricultural sustainability.

**Author Contributions:** Conceptualization, H.G.P. and T.R.C.; resources, J.M.L. and T.R.C.; writing—original draft preparation, H.G.P. and C.K.S.; writing—review and editing, H.G.P., C.K.S., J.M.L. and T.R.C.; supervision, J.M.L. and T.R.C.; project administration, H.G.P. and T.R.C.; funding acquisition, T.R.C. All authors have read and agreed to the published version of the manuscript.

**Funding:** This research received no external funding.

**Conflicts of Interest:** The authors declare no conflicts of interest.

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
