# Peer review of "Understanding Rumen Microbiology: An Overview"

_encyclopedia, doi:10.3390/encyclopedia4010013_

Round 1

Reviewer 1 Report

Comments and Suggestions for Authors

The author's have submitted an introductory review on the topic of ruminant microbiome.  The review is well prepared, structured and cited.  There are a few minor issues the author's should consider to improve readability and comprehension of this vast subject.

Minor concerns:

1.  Use of proper scientific numerical terms.  On lines 75 &85 the author's should use superscript to cite 1010, to reflect 10e10 CFU.  Some numerical terms such as 2500 on line 75 should be shown correctly as 2,500.

2.  In section 2, the author's present various parts of the microbiome as either percent of population and others as percent biomass.  It would be useful to use either or both terms with the correct numbers and proper citations.  For instance, bacteria are stated to be 70-80% of the microbial population (Line 80).  However, for the other microbe populations % of population figures are not provided.  A table which depicts percents of population and or percent of biomass may be useful.

3.  On line 135, after the term "fermentable starch is" seems to be missing a descriptive term such as "present".  Please correct this for meaning/readability.

4.  Citations in the paper are inconsistently bolded.  Please follow the style guidlines to conform to the journal style for citations. 

5.  Section 3.2 appears incomplete.  It appears the author's intent for this section would be to summarize unique challenges and methods for enumeration of bacterial populations.  Please review this section to determine if additional narrative would benefit the readers of the review.

6.  Section 5 (Conclusions) could benefit from additional comments regarding gaps in the study of ruminant microbiome studies.  The author's should consider addition of reflective comments on future studies.

Author Response

Line or Section #

Reviewers Comment

Edits made

75 & 85

Use of proper scientific numerical terms.  On lines 75 & 85 the author's should use superscript to cite 1010, to reflect 10e10 CFU.  Some numerical terms such as 2500 on line 75 should be shown correctly as 2,500.

Thank you for the valuable constructive criticism. Changes were made as suggested.

135

 after the term "fermentable starch is" seems to be missing a descriptive term such as "present". Please correct this for meaning/readability.

Thank you for catching that and providing constructive feedback. Changes were made as suggested

Section 2 – Lines #80, 106, 115

The author's present various parts of the microbiome as either percent of population and others as percent biomass.  It would be useful to use either or both terms with the correct numbers and proper citations.  For instance, bacteria are stated to be 70-80% of the microbial population (Line 80).  However, for the other microbe populations % of population figures are not provided.  A table which depicts percents of population and or percent of biomass may be useful.

While we agree with the reviewer in it would be nice, and we created that very kind of table; unfortunately because these estimates were made over a 30 year period using very different approaches and results are contradictory and appear even more confusing.  We can add that as needed, but it seems better to not add that in.   But thank you for the amazing feedback on this issue, you’re right.

Section 3.2

Section appears incomplete.  It appears the author's intent for this section would be to summarize unique challenges and methods for enumeration of bacterial populations.  Please review this section to determine if additional narrative would benefit the readers of the review.

Thank you for your valuable feedback. Section was reviewed and minor edits made.

Section 5 (Conclusions)

Could benefit from additional comments regarding gaps in the study of ruminant microbiome studies.  The author's should consider addition of reflective comments on future studies.

Thank you for the valuable constructive criticism. Changes were made as suggested for the improvement of the conclusion.

General

Citations in the paper are inconsistently bolded.  Please follow the style guidelines to conform to the journal style for citations. 

Thank you for pointing that out. Changes were made as suggested

Reviewer 2 Report

Comments and Suggestions for Authors

The review article is written on rumen microbiology which is of prime importance in animal health and production. The article provides an ample and significant information regarding the subject of rumen microbiology. 

The article is well written and seems sound for publication in the journal.

Author Response

Thank you for these kind comments

Reviewer 3 Report

Comments and Suggestions for Authors

General comment

Checked subscript and superscript in all manuscript.

Specific comment

L134: Changed to “Rumen pH is typically in the 6-7…..”

Grünberg, W., & Constable, P. D. (2008). Function and dysfunction of the ruminant forestomach. In Current veterinary therapy: food animal practice (pp. 12-19). Elsevier.

Comments on the Quality of English Language

General comment

Checked subscript and superscript in all manuscript.

Author Response

Reviewer 3 Edits:

Line or Section #

Reviewers Comment

Edits made

135

 Changed to “Rumen pH is typically in the 6-7…..”

Thank you for the valuable constructive criticism. Changes were made as suggested and reflected in citations.

General

Check subscript and superscript in all manuscript.

Thank you for the valuable constructive criticism. Changes were made as suggested.

Reviewer 4 Report

Comments and Suggestions for Authors

In my perspective this manuscript is adequate for an Entry, but authors may add some more detail about the importance of the rumen protozoa community for rumen metabolism and for animal health and nutritional quality of final products (eg. meat fatty acids profile). Also, in section 3, authors should also refer the use of 18S rRNA sequencing,  to characterize the rumen eukariotic communities (protozoa an fungi) and the optical microscopy analysis as the golden technique to study the rumen ciliate protozoa. Also a section about the factors that influence rumen microbiota communities may be added, because authors only refer rumen pH and as we known, there are many factors that influence rumen microorganisms as animal genetics background, dietary components, age, physiological status etc.  

Author Response

Reviewer 4 Edits:

Line or Section #

Reviewers Comment

Edits made

Section 3 - #106-111

Authors may add some more detail about: Rumen protozoa community for rumen metabolism and for animal health and nutritional quality of final products (eg. meat fatty acids profile) as well as 18S rRNA sequencing, to characterize the rumen eukariotic communities (protozoa an fungi) and the optical microscopy analysis as the golden technique to study the rumen ciliate protozoa

Thank you for the valuable constructive criticism. Some changes were incorporated as suggested.

Section 3 - #142

Authors may add some more detail about factors that influence rumen microbiota communities

Thank you for the constructive criticism. Some changes were incorporated as suggested.

Round 2

Reviewer 1 Report

Comments and Suggestions for Authors

Thank for your consideration and thoughtful corrections.

Reviewer 4 Report

Comments and Suggestions for Authors

I thank you for contributing to improving the scientific quality of the manuscript and for the authors considering my suggestions.

Good work.